# Transactivation of miR-202-5p by Steroidogenic Factor 1 (SF1) Induces Apoptosis in Goat Granulosa Cells by Targeting TGFβR2

**DOI:** 10.3390/cells9020445

**Published:** 2020-02-14

**Authors:** Qiang Ding, Miaohan Jin, Yaoyue Wang, Jiao Liu, Peter Kalds, Ying Wang, Yuxin Yang, Xiaolong Wang, Yulin Chen

**Affiliations:** Key Laboratory of Animal Genetics, Breeding and Reproduction of Shaanxi Province, College of Animal Science and Technology, Northwest A&F University, Yangling 712100, Shaanxi, China; dingqiang@nwafu.edu.cn (Q.D.); jinmiaohan@nwafu.edu.cn (M.J.); liujiaodk@163.com (J.L.); peterkalds@nwafu.edu.cn (P.K.); nd2013wang@163.com (Y.W.); yangyuxin2002@126.com (Y.Y.)

**Keywords:** miR-202-5p, *TGFβR2*, TGF-β signaling pathway, goat granulosa cell, cell apoptosis, SF1

## Abstract

MicroRNAs play key roles during ovary development, with emerging evidence suggesting that miR-202-5p is specifically expressed in female animal gonads. Granulosa cells (GCs) are somatic cells that are closely related to the development of female gametes in mammalian ovaries. However, the biological roles of miR-202-5p in GCs remain unknown. Here, we show that miR-202-5p is specifically expressed in GCs and accumulates in extracellular vesicles (EVs) from large growth follicles in goat ovaries. In vitro assays showed that miR-202-5p induced apoptosis and suppressed the proliferation of goat GCs. We further revealed that miR-202-5p is a functional miRNA that targets the transforming growth factor-beta type II receptor (*TGFβR2*). MiR-202-5p attenuated TGF-β/SMAD signaling through the degradation of TGFβR2 at both the mRNA and protein level, decreasing p-SMAD3 levels in GCs. Moreover, we verified that steroidogenic factor 1 (SF1) is a transcriptional factor that binds to the promoters of miR-202 and cytochrome P450 family 19 subfamily A member 1 (*CYP19A1*) through luciferase reporter and chromatin immunoprecipitation (ChIP) assays. That contributed to positive correlation between miR-202-5p and *CYP19A1* expression and estradiol (E2) release. Furthermore, SF1 repressed TGFβR2 and p-SMAD3 levels in GCs through the transactivation of miR-202-5p. Taken together, these results suggest a mechanism by which miR-202-5p regulates canonical TGF-β/SMAD signaling through targeting *TGFβR2* in GCs. This provides insight into the transcriptional regulation of *miR-202* and *CYP19A1* during goat ovarian follicular development.

## 1. Introduction

Female fecundity is closely related to follicular development in the ovaries of mammals. Primordial oocytes in the ovaries gradually increase in size ultimately forming mature oocytes and ovulate. However, less than 1% of the oocytes can cause ovulation [1,2,3] as most are wrapped in atretic follicles and are unable to ovulate. Accumulating evidence suggests that the initiation of follicular atresia is mainly due to granulosa cell (GC) apoptosis (programmed cell death) [4,5]. Ovarian endocrine factors have been extensively studied in GCs apoptosis, the regulation by small noncoding RNAs (microRNAs) has attracted more attention in recent years.

microRNAs (miRNAs) are ~22 nucleotides in length, noncoding RNAs post-transcriptionally regulate gene expression through binding to partially complementary sites on the 3′-untranslated region (3’-UTR) of target mRNAs, resulting in mRNA degradation or inhibiting gene transcription. Dynamic changes in the levels of miRNA expression occur during ovarian development with different miRNA profiles existing [6,7]. To date, several miRNAs have been identified as regulators of GCs apoptosis in domestic animals, including miR-92a [8], miR-1275 [9], and let-7g [10]. Despite the identification of many candidate miRNAs, their biological functions in the ovaries remains unclear. MiRNA-mediated GCs apoptosis therefore requires further exploration.

In this study, we selected miR-202-5p for further studies based on its significant enrichment in the extracellular vesicles from follicular fluid in goat ovarian follicles. MiR-202-5p is a germ plasm unique miRNA that is abundantly enriched in the gonads of some mammals and lower vertebrates. In mouse, miR-202-5p is specifically expressed in the Sertoli cells of embryonic gonads [11] and in spermatogonial stem cells (SSCs) in the testis [12]. In human testis, miR-202-5p is a germ cell-dependent expressing miRNA in Sertoli cells [13], indicating its role in spermatogenesis. Moreover, miR-202 is conservatively expressed in the germ cells of lower vertebrates, such as frogs [14,15] and fish. The expression of miR-202-5p gradually increases during zebrafish oocyte growth and peaks in mature oocytes [16]. *MiR-202* silencing in medaka fish impairs early folliculogenesis, subsequently reducing the number of eggs leading to abnormal eggs that cannot be fertilized, leading to low-fecundity [17], implying a role for miR-202-5p during oogenesis. Previously studies reported that miR-202-5p is typically abundant in large animal ovarian follicles [6,18,19], but its roles remained elusive. In this study, we investigated the expression of miR-202-5p in goat GCs of different follicle sizes and further demonstrate its biological functions. In addition, we confirmed a transcriptional target of miR-202 and reveal its specific roles in ovarian follicles. Therefore, we have identified that miR-202-5p might be a potential marker for ovulation in goats.

## 2. Materials and Methods

### 2.1. Follicular Separation, Cell Isolation, and Culture

Adult goat ovaries were collected at a slaughterhouse (Baoji, Shaanxi, China) and transported to the laboratory at ~4 °C in 0.9% physiological saline solution. Ovaries were selected based on the presence of at least one obvious large follicular follicle at the ovarian surface.

Goat ovarian follicles were dissected and sorted according to their diameter (small follicles <3 mm and large follicles >5 mm). GCs, cumulus oocyte complexes (COCs), and naked-oocytes were squeezed from the 2–5 mm diameter goat follicles of mature ovaries using 1 mL syringes. COCs and naked oocytes were sorted using mouth pipettes. GCs were seeded into 24-well culture plates at a density of 1 × 10^6^ cells and cultured in a serum-free medium containing sodium bicarbonate (10 mmol/L), sodium selenite (4 ng/mL), bovine serum albumin (BSA) (0.1%; Sigma-Aldrich, Saint Louis, MO, USA), penicillin (100 U/mL), streptomycin (100 μg/mL), 1 × ITS-A (Insulin-Transferrin-Selenium-Sodium Pyruvate (ITS-A) (100×), 51300044, ThermoFisher Scientific, Green Island, NY, USA), nonessential amino acid mix (1.1 mmol/L), androstenedione (10^−7^ M at start of culture and 10^−6^ M at each medium change) and bovine follicle-stimulating hormone (FSH) (10 ng/mL starting on day 2; AFP5346D; National Hormone and Peptide Program, Torrance, CA, USA). GCs were maintained in 5% CO_2_ at 37 °C for 4 d prior to treatment and replaced with a 70% medium every 48 h. COCs and naked oocytes were collected and frozen at −70 °C.

### 2.2. RNA Isolation and qPCR

Total RNAs were extracted from ovarian follicles, COCs, naked oocytes, cultured GCs, and follicles fluid EVs using RNAiso (TaKaRa, Dalian, China) based on the manufacturer’s protocols. Total RNAs were reverse transcribed using First Strand cDNA synthesis kits (ThermoFisher Scientific, USA). Using specific miRNA reverse primers, the stem-loop method was used for miRNA synthesis. Relative mRNA expression was measured with qPCR using SYBR Premix EX Taq II (TaKaRa, Dalian, China). PCRs were performed in triplicate. GAPDH was used as internal control for mRNA, with U6 used for miRNA. The 2^−∆∆Ct^ method was used to normalize the relative expression levels of the target genes [20]. Primers sequences are shown in Appendix A.

### 2.3. Small RNAs Sequencing of FF-EVs

Total RNA was evaluated with an Agilent 2100 Bioanalyzer (Agilent Technologies, Santa Clara, CA, USA). RNA concentrations were measured using Qubit^®^ RNA assay kits in a Qubit^®^ 2.0 Flurometer (Life Technologies, Eugene, OR, USA). A total of 20 ng of small RNA per exosome was used for the construction of small RNA sequencing libraries. Each sample library was built following the manufacturer’s recommendations of NEBNext^®^ Multiplex Small RNA Library Prep Set for Illumina^®^ (NEB, Ipswich, MA, USA). All libraries were sequenced on an Illumina Hiseq 2500 platform and 50 bp single-end reads were generated. MiRNA expression levels were estimated by transcript per million (TPM) as described [21]: Normalized expression = mapped read count/total reads × 1,000,000. Differential miRNA expression across the two EVs groups were performed using the DESeq R package [22]. *p*-values were adjusted using q-values; q-value < 0.01, and |log2 (fold change)| > 1 were set as the threshold for significantly differential expression.

### 2.4. Plasmids and siRNAs

PcDNA3.1(+)-neo, pGL3-basic, pRL-TK, and psiCheck^TM^-2 vectors were kindly provided by Wenxian Zeng’s laboratory (College of Animal Science and Technology, Northwest A & F University, China). The promoter region of *CYP19A1* or miR-202 were amplified from the total DNA of goat GCs and subcloned into the pGL3-basic for luciferase assays. Partial sequences of the TGFβR2 3′-UTR regions (containing miR-202-5p binding sites) were amplified from the cDNA of goat GCs and subcloned into psiCheck^TM^-2. Overlap PCRs were performed to generate mutant plasmids and successful mutations were identified by Sanger sequencing. For overexpression studies, CDS fragments of SF1 and TGFβR2 were amplified from GC cDNA and subcloned into pcDNA3.1(+)-neo. MiR-202-5p mimics, miR-202-5p inhibitors, mimic NC, inhibitor NC, SF1-siRNA, TGFβR2-siRNA, and siRNA-NC were chemically synthesized by Shanghai GenePharma (Shanghai, China). SiRNA sequences are listed in Appendix A.

### 2.5. Cell Transfection

GCs were cultured in a fresh medium prior to DNA or RNA transfection. MiRNAs, plasmids, and siRNAs were transfected into cells using Lipofectamine^®^ 3000 (ThermoFisher Scientific, CA, USA) according to the manufacturer’s instructions. Cells were harvested for RNA or protein extraction 72 h post-transfection.

### 2.6. Estradiol ELISA Assay

Follicular fluid (FF) was isolated from the follicles. Samples were centrifuged for 10 min at 3000 rpm to remove cells and cellular debris. The concentration of estradiol in the FF was measured using a Goat estrogen ELISA kit (Shanghai Meilian Biotechnology Company, Shanghai, China) according to the manufacturer’s protocols.

### 2.7. Western Blotting (WB)

The EVs of FF were purified by differential ultracentrifugation and directly suspended in a RIPA buffer. Treated cells were washed three times with cold PBS and lysed in a 100 μL/well RIPA buffer containing 1 mM PMSF (Solarbio, P0100, Beijing, China) and phosphatase inhibitors (Roche, 4906845001, Penzberg, Germany). Lysed cells were centrifuged and supernatants were transferred into fresh 1.5 mL centrifuge tubes. Total protein concentrations were determined by the BCA assay (Vazyme, Nanjing, China). All samples were mixed with a 4 × SDS loading buffer. After boiling, each sample (20 μg) was subjected to 12% SDS-PAGE in running buffer at 90 V for 2 h. Proteins were electro-transferred to PVDF membranes at 260 mA for 2 h. PVDF membranes were incubated for 1 h at room temperature in TBST containing 5% skimmed milk power. Membranes were incubated at 37 °C for 2 h or 4 °C overnight following primary antibodies [anti-TGFBR2 1:1000, Cat. No. 66636-1-Ig; anti-SMAD4, 1:1000, Cat. No. 10231-1-AP; anti-NR5A1 (SF1), 1:200, Cat. No. 18658-1-AP; anti-Beta actin, 1:2000, Cat. No.20536-1-AP (Proteintech, Wuhan, China); anti-SMAD3 (1:1000, BA4559, BOSTER, Wuhan, China), and anti-phospho-SMAD3 (Phospho-S423/S425) antibodies (1:800, 13370, SAB)]. After washing three times with TBST, membranes were labeled for 1 h at room temperature with HRP-conjugated Affinipure Goat Anti-Rat IgG (H + L) (1:2000, Cat. No. SA00001-15, Proteintech, Wuhan, China) antibodies. Protein bands were detected using a chemiluminescence reagent (ECL; Millipore, Billerica, MA, USA) and visualized on an imaging system. Protein bands were analyzed using ImageJ software (ImageJ 1.48V).

### 2.8. Chromatin Immunoprecipitation (ChIP)-PCR Assay

ChIP assays were performed using ChIP assay kits (Beyotime, P2078, Shanghai, China) according to the manufacturer’s protocol. Cells were treated in 1% methanol at 37 °C for 10 min and glycine was added to stop the reaction. After washing with cold PBS (×3), cells were lysed with SDS lysis buffer and ultrasound fragmentation assays were used to obtain 300–500 bp DNA fragments. The ChIP dilution buffer was added to produce ultrasound products. SF1-antibodies (Rabbit mAb #12800, CST) or control IgG were added at 4 °C overnight. DNA was purified using DNA extraction kits. Primers for ChIP assays are listed in Appendix A.

### 2.9. Dual Luciferase Assays

After 36 h transfection, HEK293T cells were washed in PBS and lysed. Relative luciferase activity was measured using dual luciferase assay kits according to the manufacturer’s protocols (E1910, Promega, Madison, WI, USA).

### 2.10. Apoptosis Assays

GC apoptosis was assessed using Annexin V-FITC/PI apoptosis detection kits (A211, vazyme, Nanjing, China) according to the manufacturer’s protocols. Fluorescence activated cell sorting (FACS) was used to detect GC apoptosis. Early apoptotic cells were stained with Annexin V-FITC and late apoptotic cells were stained with propidium iodide (PI). The apoptotic rates of GCs were calculated and analyzed using Flowjo software (v7.6, Stanford University, Stanford, CA, USA).

### 2.11. Cell Proliferation Assays

GC proliferation assays were performed using BeyoClick EdU cell proliferation kits (Beyotime, Shanghai, China). Cells were cultured in media containing EdU-488 for 12 h and fixed with 4% paraformaldehyde for 15 min at room temperature. Cells were permeabilized in 0.3% Triton X-100 for 15 min at room temperature and click additive solution was used to detect EdU. Hoechst 33342 (5 μg/mL) was used to stain cell nuclei. EdU-488 positive cells were detected on an inverted fluorescence microscope (AMG EVOS, Mill Creek, WA, USA).

### 2.12. Statistical Analysis

Experiments for each group were repeated at least three times, data are the means ± SEM. The GraphPad Prism 6 Software was used to perform statistical analysis. A Spearman’s rank correlation coefficient was used in correlation analysis. A two-tailed Student’s *t*-test was used for groups comparisons. Three or more groups were compared using one-way analysis of variance and Turkey’s test. *p*-value < 0.05 and *p*-value < 0.01 were considered as significant and extremely significant differences.

## 3. Results

### 3.1. miR-202-5p Was Highly Expressed in Growing Follicles

In previous studies, EVs were isolated from goat ovary follicular fluid and EVs-miRNAs were also identified. Heatmap revealed differential expression profiles of EV-miRNAs from small follicular follicles (SFF) and large follicular follicles (LFF) (Figure 1A), that revealed alterations in miRNA levels in the EVs during ovarian follicle development. We found that miR-202-5p was highly expressed in LFF-EVs compared to SFF-EVs (Figure 1B). To further explore the expression of miR-202-5p in ovary follicles, ovarian cells were separated and sorted based on follicle diameter. Realtime-PCR analysis showed that miR-202-5p was highly expressed in GCs in large growth follicles (Figure 1C). MiR-202-5p was also enriched in cumulus cells as opposed to naked-oocytes (Figure 1C). These results suggest that miR-202-5p is functional in the somatic cells of goat ovaries during follicular growth and may regulate oocyte maturation.

### 3.2. miR-202-5p Induces GC Apoptosis and Suppresses GC Proliferation In Vitro

To further investigate the biological functions of miR-202-5p in GCs, miR-202-5p mimics or inhibitors were transfected into cultured GCs in vitro. FACS analysis showed that the rates of GC apoptosis increased after transfection with miR-202-5p mimics compared to NC, whilst inhibiting the expression of miR-202-5p in GCs decreased cell apoptosis rates compared to NC transfections (Figure 2A). In addition, RT-PCR analysis showed that the pro-apoptotic gene *BAX* was highly expressed in miR-202-5p overexpressed GCs compared to cells transfected with NC mimics (Figure 2B and Appendix A). Inhibiting miR-202-5p downregulated *BAX* expression whilst *Bcl2* expression increased (Figure 2C and Appendix A). Furthermore, the effects of miR-202-5p on GCs proliferation was verified using EdU assays. Fluorescence analysis suggested that the overexpression of miR-202-5p suppressed GC proliferation 48 h post transfection (Figure 2D). qPCRs further showed that miR-202-5p inhibits the expression of *Cyclin D*, *Notch2,* and *PCNA* which were associated with cell proliferation (Figure 2E). In contrast, the inhibition of miR-202-5p promoted GC proliferation in vitro (Figure 2D) and increased the relative genes mRNA expression (Figure 2F). Taken together, these results indicate that miR-202-5p induces cell apoptosis and suppresses the proliferation of goat GCs in vitro.

### 3.3. miR-202-5p Binds and Downregulates TGFβR2 mRNA Expression in GCs

To further understand the function of miR-202-5p, we evaluated its cellular targets. Based on three miRNA target databases (miRTarBase, miRDB, and TargetScan7), nine genes were considered (Figure 3A). Among these, *TGFβR1* and *TGFβR2* were most relevant to follicles development. In addition, bioinformatics showed that miR-202-5p targets *TGFβR2* in vertebrates (Figure 3B) and identified the existence of three multiple putative miR-202-5p binding sites within the *TGFβR2* 3′-UTR of goats (Figure 3B). To further confirm that miR-202-5p directly binds to *TGFβR2* 3′-UTR, luciferase reporter vectors expressing three miR-202-5p binding sites in the *TGFβR2* 3′-UTR or mutated binding sites were transfected and relative luciferase activity was assessed (Figure 3C). The relative luciferase activity of all three wildtype reporters was significantly reduced in cells expressing miR-202-5p mimics compared to NC mimics, whilst mutant 3′-UTR constructs remained unaffected (Figure 3D). To investigate whether miR-202-5p regulates the expression of *TGFβR2* in goat GCs, miR-202-5p mimics or inhibitors were transfected into cultured goat GCs. The overexpression of miR-202-5p significantly decreased both mRNA and protein expression of *TGFβR2* (Figure 3E,F), whilst inhibiting miR-202-5p had no significant effects on *TGFβR2* expression (Figure 3E,F). *TGFβR1* was also predicted as a target of miR-202-5p. However, neither miR-202-5p overexpression nor silencing in goat GCs altered the levels of *TGFβR1* (Figure 3E). Taken together, these results demonstrate that miR-202-5p directly regulates *TGFβR2* expression in goat GCs through binding to the *TGFβR2* 3′-UTR.

### 3.4. miR-202-5p Inhibits TGF-β/SMAD Signaling in GCs

*TGFβR2* is a canonical reporter of the TGF-β signaling pathway that regulates downstream SMAD3 and p-SMAD3 expression levels. Next, we investigated whether miR-202-5p to influence GC apoptosis and/or TGF-β/SMAD signaling via targets *TGFβR2*. SMAD3 and p-SMAD3 protein levels were detected in GCs cotransfected with pcDNA3.1-*TGFβR2* and miR-202-5p mimics. The results showed that TGFβR2 could rescue the apoptosis of GCs mediated by miR-202-5p (Figure 4A), whilst GCs transfected with *TGFβR2*-siRNA showed increased rates of apoptosis (Figure 4B). WB analysis of the proteins downstream of TGF-β showed that overexpressing miR-202-5p dramatically decreased p-SMAD3 (Phospho-S423/S425) levels in GCs. TGFβR2 overexpression in GCs prevented the loss of p-SMAD3 and increased p-SMAD3/SMAD3 levels (Figure 4C). Reducing the expression of TGFβR2 inhibited the miR-202-5p inhibitor-mediated enhancement of p-SMAD3 and decreased p-SMAD3/SMAD3 levels (Figure 4D). These findings suggest that miR-202-5p downregulates *TGFβR2* expression to modulate TGF-β/SMAD signaling in goat GCs.

### 3.5. SF1 Promotes miR-202 and CYP19A1 Expression through Binding to Their Promoter Regions

To understand the mechanism(s) through which miR-202-5p is upregulated during follicular development, we focused on the transcriptional regulation and promoter regions. We noted that SF1 acts as a transcriptional factor for both *miR-202* and *CYP19A1* in several studies [11,23,24]. To define the relationship between SF1 and *miR-202* and *CYP19A1* in goat ovaries. GCs and FF were separated for RNA extraction and E2 detection, respectively. A total of 22 large follicular follicles were dissected from goat ovaries and semi-quantitative RT-PCR was used to assess gene expression. The relative mRNA level of SF1 in the large follicles significantly and positively correlated with both *miR-202-5p* (*r* = 0.5626, *p* = 0.0018) (Figure 5A, left) and *CYP19A1* (*r* = 0.5754, *p* = 0.0099) (Figure 5A, middle) expression. As predicted, *CYP19A1* expression positively correlated with E2 levels (*r* = 0.7941, *p* < 0.0001) (Figure 5B, left), whilst SF1 also significantly and positively correlated with E2 release in goat GCs (*r* = 0.6485, *p* = 0.0011) (Figure 5B, middle). We further observed a positive correlation between miR-202-5p expression and *CYP19A1* expression (*r* = 0.5632, *p* = 0.0121) (Figure 5A, right) and E2 release (*r* = 0.452, *p* = 0.0454) (Figure 5B, right) in goat ovarian follicles. These results suggested that miR-202-5p is a potential marker for follicle development.

To further confirm the ability of SF1 to regulate the transcription of *miR-202* and *CYP19A1*, we cultured GCs and transfected pcDNA3.1-SF1 or SF1-siRNA *in vitro*. The mRNA levels of miR-202-5p were significantly higher in GCs transfected with pcDNA3.1-SF1 compared to pcDNA3.1 and were suppressed following SF1 silencing in GCs (Figure 5C). As expected, *CYP19A1* expression was enhanced by the overexpression of SF1, and downregulated following SF1 silencing in cultured GCs (Figure 5D). Next, we focused on the transcriptional regulation of *miR-202* and *CYP19A1* and their promoter regions. We obtained the sequences of these promoter regions containing nuclear receptor elements (NRE) of goat *CYP19A1* (CCAAGGTCA, −241~−249 nt) and *miR-202* (CCAAGGTCT, −259~−267 nt). A SF1-0.1 binding sites in *miR-202* and *CYP19A1* have been reported. *CYP19A1* or pri-miR-202 promoter luciferase reporters with wildtype or mutated NREs were generated and co-transfected with pcDNA3.1-SF1 into GCs. We observed an increase in the activity of the wildtype promoter in cell expressing *CYP19A1* and *miR-202* (Figure 5E). Meanwhile, the relative luciferase activity in GCs expression of the mutated SF1 binding sites decreased compared with wildtype promoters (Figure 5E). ChIP-PCR assays were performed to explore the relationship between SF1 and *CYP19A1* or *miR-202* promoters in vivo. The immunoreactive signals of SF1 were more intense than IgG negative controls (Figure 5F). These data suggest that SF1 is a transcription factor for both *CYP19A1* and *miR-202.* This highlights the association of SF1 with estrogen release and the regulation of *miR-202* and *CYP19A1* expression through direct binding to their promoters in goat GCs in vivo and in vitro.

### 3.6. miR-202-5p Mediates SF1 Regulation of the Canonical TGF-β/SMAD Signaling Pathway

As SF1 enhances miR-202 expression and miR-202-5p targets *TGFβR2* in GCs, we investigated the mechanisms through which SF1 regulates canonical TGF-β/SMAD signaling in GCs. qPCR confirmed that SF1 overexpression decreased the mRNA expression of *TGFβR2* in GCs (Figure 6A). Similarly, WB showed that SF1 decreased TGFβR2 protein levels in cultured GCs, confirming its ability to negatively regulate TGFβR2 expression (Figure 6B). To further confirm these findings, we cotransfected miR-202-5p inhibitors with pcDNA3.1-SF1 and miR-202-5p mimics with SF1 specific-siRNAs into goat GCs. The results showed that miR-202-5p inhibitor rescued the SF1-mediated decrease in TGFβR2 and p-SMAD3 (Phospho-S423/S425) levels (Figure 6B). In contrast, the overexpression of miR-202-5p inhibited the SF1 specific-siRNA increase in TGFβR2 and p-SMAD3 (Phospho-S423/S425) protein expression (Figure 6C). These results demonstrated that SF1 negatively regulates TGF-β/SMAD signaling through miR-202-5p.

## 4. Discussion

MiRNAs regulate an array of physiological processes through the post-transcriptional regulation of gene expression. In this study, we identified miR-202-5p as a highly expressed miRNA in the GCs of goat large ovary follicles and further explored its role in GC apoptosis via TGF-β/SMAD signaling. We further identified SF1 as a common transcriptional factor that promotes both *miR-202* and *CYP19A1* expression in GCs.

Our data showed that miR-202-5p was highly expressed in goat GCs during ovarian follicle growth, suggesting its functionality in pre-ovulation follicles. In ruminants, miR-202 shows restricted expression in bovine ovaries, whilst its expression increased in large healthy follicles, particularly in GCs [6]. Furthermore, we found that miR-202-5p positively correlated with *CYP19A1* expression in goat ovarian follicles. *CYP19A1* catalyzes the conversion of androgens to estrogens, thought to be the rate-limiting step during estrogen biosynthesis [23]. This showed a very important role in animal ovarian development. *CYP19A1* expression was also restricted to large antral healthy follicles and pre-ovulatory follicles in mature animal ovaries [6,24]. This indicated a relationship between miR-202-5p and *CYP19A1* and suggested a potential role for miR-202-5p in E2 release and follicle development. The expression of miR-202-5p in the GCs of larger follicles was higher than that of small follicles and naked-oocytes in goat ovaries. MiR-202 could therefore be considered as a candidate miRNA with potentially important functions in animal reproduction.

MiRNAs function by binding to the mRNA of target genes. In this study, *TGFβR2* was identified as a functional target of miR-202-5p and overexpressing miR-202-5p induced GC apoptosis through inactivating TGF-β/SMAD signaling through the degradation of *TGFβR2* transcription. *TGFβR2* is a transmembrane receptor of the TGF-β/SMAD signaling that induces SMAD2/3 activity [25]. The downregulation of phospho-SMAD2/3 is pro-apoptotic. Herein, the expression of p-SMAD3 was significantly reduced in goat GCs following treatment with miR-202-5p mimics or *TGFβR2*-siRNA. SMAD3 activity could be rescued by the overexpression of *TGFβR2*. This indicated that miR-202-5p suppresses TGF-β/SMAD signaling in apoptotic GCs. However, we observed no significant changes in the levels of *TGFβR2* and p-SMAD3 in GCs treated with miR-202-5p inhibitors. It is likely that the low levels of endogenous miR-202-5p in GCs in vitro was unaffected by *TGFβR2* expression. Studies have demonstrated the ability of several miRNAs to suppress TGFβR2 expression and induce cell apoptosis via inactivating *TGFβR2*-dependent TGF-β/SMAD signaling in GCs [26,27,28]. In mammalian ovaries, miRNAs that regulate members of the TGF-β signaling pathway have been identified in atretic follicles. For example, miR-224 and miR-26b regulate GC apoptosis through their effects on SMAD4, a key factor of the canonical TGF-β signaling [29,30]. miR-425 downregulates TGFβR2, leading to GCs apoptosis in porcine ovaries [27]. Studies on miRNAs and TGF-β signaling pathways in ovarian follicles highlight its importance in follicle atresia. This study provides new evidence of the miRNA regulation of goat GC genes and indicates that miR-202-5p regulates TGF-β signaling.

To our knowledge, TGFBR2 is the only TGFβ receptor able to bind all the TGFβ ligands and elicit functional signaling to cells. *TGFβR2* activates TGF/activin and BMP/GDF pathways into cells through SMAD2/3 and SMAD1/5/8, respectively [31]. Furthermore, in vitro, FSH increases the levels of TGFB1, TGFBR2, SMAD2, SMAD3, and SMAD4 in mouse GCs. The levels of phospho-SMAD3, but not phospho-SMAD2, also increased in response to FSH, demonstrating the continual activation of TGFβ signaling [32]. SMAD3 in GCs enhances the upregulation of *TGFβR2* expression to promote cell proliferation through facilitating the effects of TGFβ on GCs [33]. Here, the apoptosis of goat GCs was enhanced by the overexpression of miR-202-5p and *TGFβR2* specific small interfering RNAs, whilst phosphor-SMAD3 was also attenuated. This suggests that TGFβ signaling was mainly activated by mediating TGFβR2 in goat GCs. In ovarian cells, *TGFβR2* is expressed throughout all stages of follicular development, including primary follicles in the embryonic phase, pre-ovulation follicles, and cells of the luteal corpus [34,35]. Moreover, TGFβR2-specific depletion in GCs using FSHR-Cre mice led to a weak maintenance of oocyte meiotic arrest within large antral follicles. TGFβR2 depletion also impaired follicle development, ovulation, and female fertility [32].

MiR-202 is an intronic RNA that participates in multiple cellular physiological processes, including cell growth, apoptosis, migration, and invasion [36,37,38]. The expression of miR-202-5p is regulated by SF1 or SRY (Sex-Determining Region Y)-Box 9 (SOX9) during mouse testis differentiation [11]. In this study, we observed a positive relationship between *miR-202-5p* and *SF1* and *CYP19A1* in goat follicles. Characterization of the putative *miR-202* promoter in vitro suggested that SF1 transactivates *miR-202* at specific genomic regions, with mutations of specific SF1 binding sites ameliorating this transactivation. In addition, we found that SF1 binds to the *CYP19A1* promoter region to enhance *CYP19A1* expression and E2 concentrations in goat. These results fully describe the positive correlation between miR-202-5p and *CYP19A1* gene in goat GCs. *CYP19A1* contains multiple promoters, whilst the ovary-specific expression patterns of aromatase are controlled by type II promoters (PII) that reside within the immediate 5′ flanking regions of the translational start site [39]. SF1 was shown to regulate *CYP19A1* transcription through its binding to the nuclear receptor motifs within the PII promoter region [40,41]. In this study, an additional SF1 motif was verified in the sequence of goat *CYP19A1* gene, also termed SF1-0.1 in mice [11]. SF1 as a key functional regulator of ovary development [42], acts at multiple levels of the reproductive axis [43]. SF1 silencing indicated its vital role during adrenal and gonadal development [44]. The ovaries of SF1-silenced mice were hypoplastic and sterile and showed reduced numbers of oocytes and lacked corpora lutea [45]. SF1 belongs to the nuclear receptor subfamily that requires a cAMP-response element-binding protein (CREB)-regulated transcription coactivator (CRTC2) and calcineurin to regulate to *CYP19A1* expression in rat ovarian GCs [46]. Moreover, TGF-β1 was confirmed to enhance SF1 (a nuclear receptor NR5A subfamily member) and LRH-1 (liver receptor homolog 1) expression in GCs treated with FSH [46]. However, in human adrenocortical cells, TGF-β1 suppresses endogenous SF1 levels and the overexpression of SMAD3 inhibits SF1 binding to the *CYP17* promotor [47]. Additionally, SF1 was required for TGF-β3-induced *CYP19A1* expression and E2 release, whilst TGF-β3 enhanced the binding of SF1 to the endogenous ovary-specific *CYP19A1* type II promoter [48]. The relationship between SF1 and TGF-β signaling is therefore complex and requires further studies in steroidogenesis cells. In addition, the overexpression SF1 could suppress TGFBR2 expression through miR-202-5p in GCs. This indicated the ability of SF1 to regulate the downstream targets of TGF-β1. Taken together, these results suggest the presence of a negative feedback loop between SF1 and TGF-β signaling mediated by miR-202-5p in GCs.

## 5. Conclusions

Our results provide an alternative regulatory mechanism of miR-202-5p mediated apoptosis in the GCs of goat ovaries and identified SF1 as a transcription factor that promotes both miR-202 and *CYP19A1* expression (Figure 7). Our findings suggest that *TGFβR2* expression was suppressed by miR-202-5p and SF1 in cultured GCs in vitro, providing a rationale for the gene-miRNA-gene axis in ovarian follicles. Collectively, these data reveal the existence of a new regulatory axis in goat ovaries through canonical TGF-β/SMAD signaling during follicular development.

## Figures and Tables

**Figure 1 cells-09-00445-f001:**
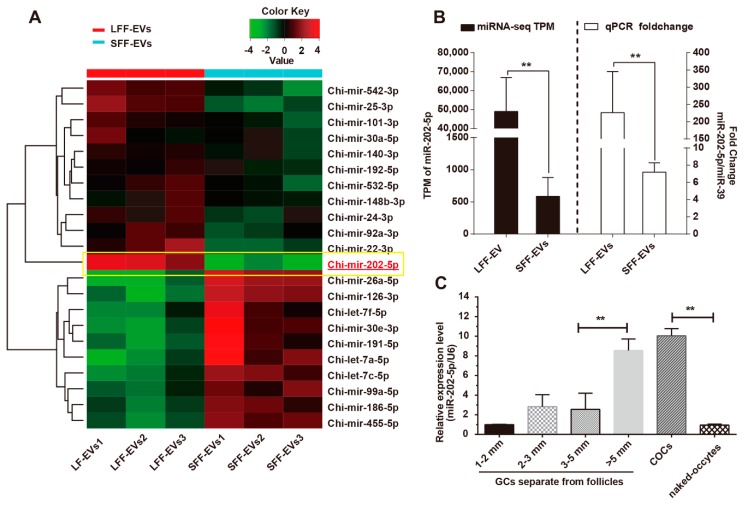
MiR-202-5p is highly expressed in large goat follicles. (**A**) Heatmap of microRNA (miRNA) expression between the large follicular fluid- extracellular vesicles (LFF-EVs) and the small follicular fluid- extracellular vesicles (SFF-EVs). (**B**) qPCR revealed the high levels of miR-202-5p in LFF-EVs, miR-39 was used as an external control. (**C**) Relative expression of miR-202-5p in granulosa cells (GCs), cumulus-oocyte-complex (COCs), and naked-oocytes. Samples are a minimum of three independent repeats. Values are normalized to U6 controls. Bars indicate the mean ± SEM of three independent replicates; ***p* < 0.01.

**Figure 2 cells-09-00445-f002:**
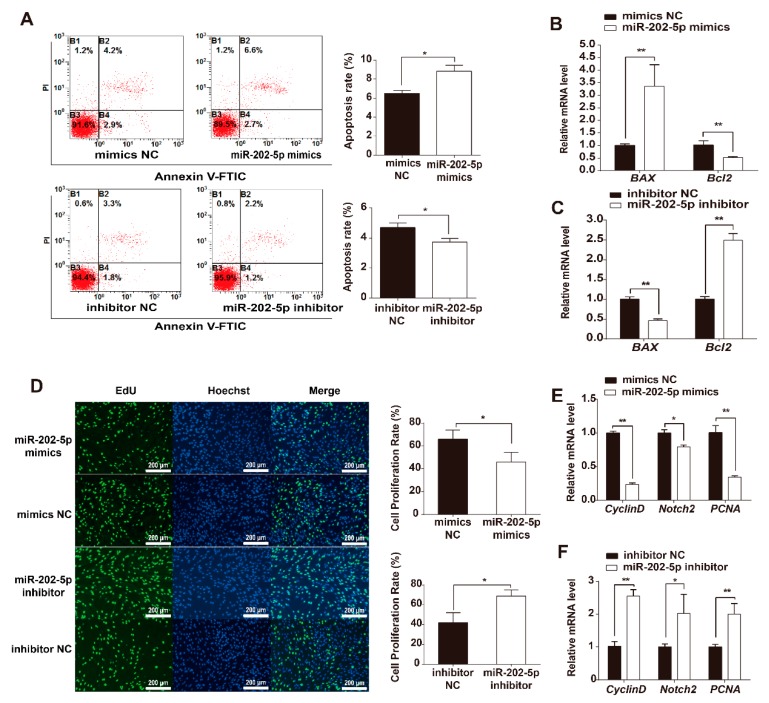
MiR-202-5p induces GC apoptosis and suppresses GC proliferation. (**A**) miR-202-5p induces cell apoptosis. GCs were treated with miR-202-5p mimics or miR-202-5p inhibitors and mimics negative control (NC) or inhibitor NC. Cell apoptosis were measured by Fluorescence activated cell sorting (FACS) and apoptotic rates were calculated. (**B**,**C**) mRNA levels of *BAX* and *Bcl2* detected by qPCR. (**D**) GC proliferation assessed via EdU assays. Cells were cultured in EdU-488 reagent after transfection with miR-202-5p mimics or miR-202-5p inhibitors and mimics NC or inhibitor NC. EdU-488 represents proliferative cells; blue Hoechst 33342 staining showing total cell numbers, scale bar = 200 μm. (**E**,**F**) mRNA levels of *CyclinD*, *Notch2,* and *PCNA* were analyzed by qPCR. All samples were measured from at least three independent experiments. *GAPDH* was used as an internal control for qPCRs. Bars indicate the mean ± SEM of three independent replicates; **p* < 0.05, ***p* < 0.01.

**Figure 3 cells-09-00445-f003:**
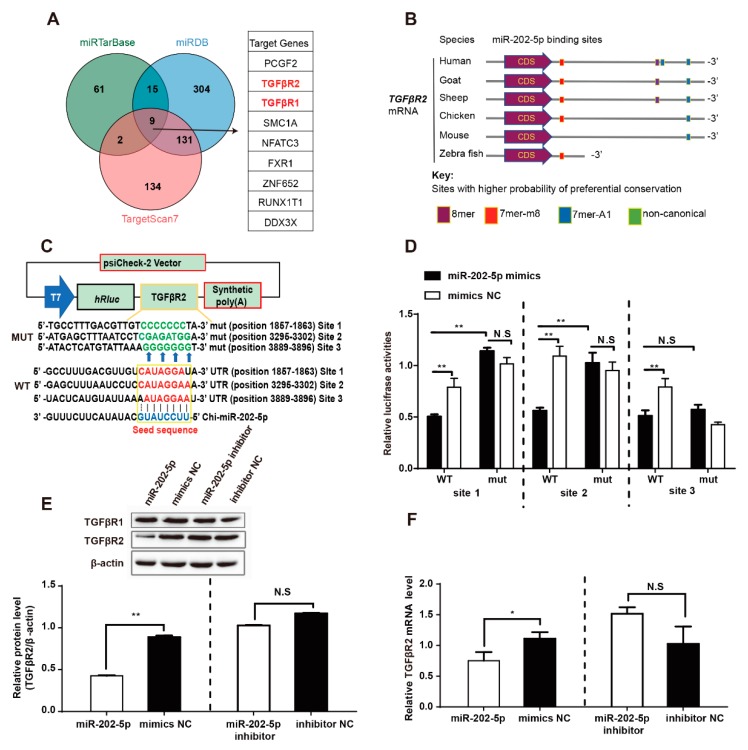
*TGFβR2* is a direct target of miR-202-5p. (**A**) Targets of miR-202-5p were predicted using three miRNA databases. Nine overlapping genes are shown. (**B**) *TGFβR2* is a target of miR-202-5p in different species, rectangles indicated the binding sites of miR-202-5p. (**C**) Schematic illustrating the design of luciferase reporters with three miR-202-5p binding sites in wildtype *TGFβR2* 3′UTR (WT) or mutant *TGFβR2* 3′UTR (mut). (**D**) *TGFβR2* 3′UTR or its mutant luciferase reporter vectors were cotransfected with miR-202-5p mimics (or negative control) into HEK293T cells. Cells were lysed 48 h post-transfection and assessed for relative luciferase activities. (**E**) Western blotting (WB) was used to analyze the expression of TGFβR1 and TGFβR2. (**F**) *TGFβR2* mRNA levels were detected in GCs transfected with miR-202-5p mimics, inhibitors, mimics NC or inhibitor NC. Experiments were performed on a minimum of three occasions. Bars indicate the mean ± SEM of three independent replicates; **p* < 0.05, ***p* < 0.01, N.S.: Not significant.

**Figure 4 cells-09-00445-f004:**
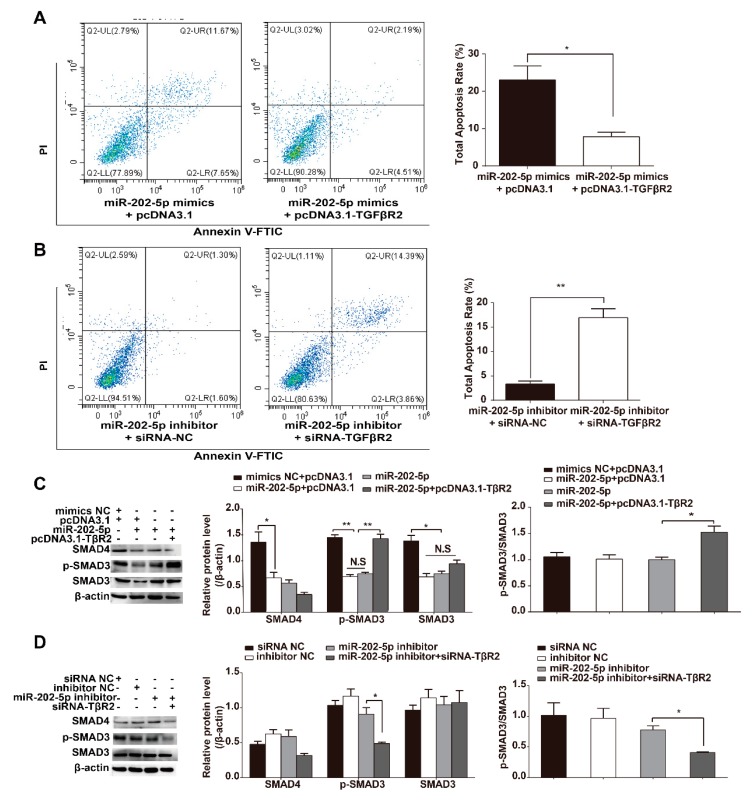
MiR-202-5p functionally targets *TGFβR2* to regulate GC apoptosis through canonical TGF-β/SMAD signaling. Overexpression of *TGFβR2* rescues cell apoptosis in response to miR-202-5p mimics (**A**). *TGFβR2* silencing induces apoptosis in GCs transfected with miR-202-5p inhibitors (**B**). Cell apoptosis was assessed via FACS ((**A**,**B**) left panels) and positive rates were calculated ((**A**,**B**) right panels). (**C**,**D**) GCs were transfected with the small interfering RNAs (siRNAs) or plasmid vectors and protein levels were analyzed by WB. GCs were cotransfected with miR-202-5p mimics and pcDNA3.1-TGFβR2 (TβR2) or miR-202-5p inhibitor and *TGFβR2*-siRNA. Relative protein levels ((**C**,**D**) middle panels) and p-SMAD3/SMAD3 ((**C**,**D**) right panel) expression were calculated. Each experiment was repeated on a minimum of three occasions. Bars indicate the mean ± SEM of three independent replicates; **p* < 0.05, ***p* < 0.01, N.S.: Not significant.

**Figure 5 cells-09-00445-f005:**
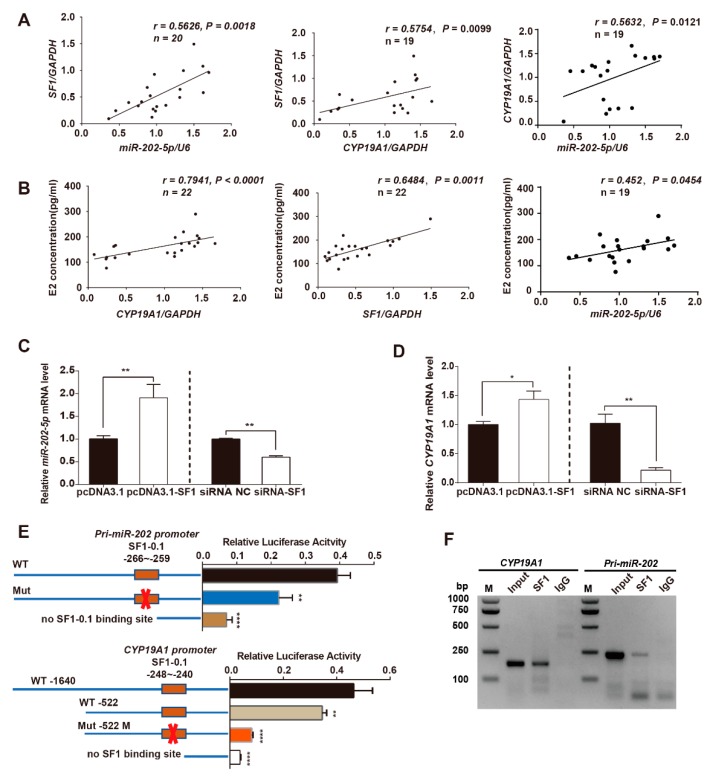
Steroidogenic Factor 1 (SF1) regulates both *miR-202-5p* and *CYP19A1* expression and is associated with estrogen release. (**A**) Correlation between SF1 and *miR-202-5p* and *CYP19A1. U6* and *GAPDH* were used as references for miRNA and mRNA expression, respectively. (**B**) Correlation between E2 in the follicular fluid and the relative mRNA expression of goat GCs. (**C**,**D**) Goat GCs were cotransfected with pcDNA3.1-SF1, pcDNA3.1 vector, siRNA-NC and SF1-siRNA, and total RNA was extracted 48 h post-transfection. mRNA levels of *miR-202-5p* and *CYP19A1* were detected by qPCR. (**E**) SF1 regulates both *miR-202* and *CYP19A1* promoter activities. HEK293T cells were cotransfected with pcDNA-3.1-SF1 and reporter vectors harboring wildtype or mutant promoters. Renilla and firefly luciferase were assessed and relative luciferase activity was calculated. (**F**) Chromatin immunoprecipitation (ChIP) assays to detect SF1 binding to *pri-miR-202* and *CYP19A1* promoters. Experiments were repeated three times. Correlation analysis was performed using Spearman’s rank correlation coefficients. Bars indicate the mean ± SEM of three independent replicates; **p* < 0.05, ***p* < 0.01, *****p* < 0.0001.

**Figure 6 cells-09-00445-f006:**
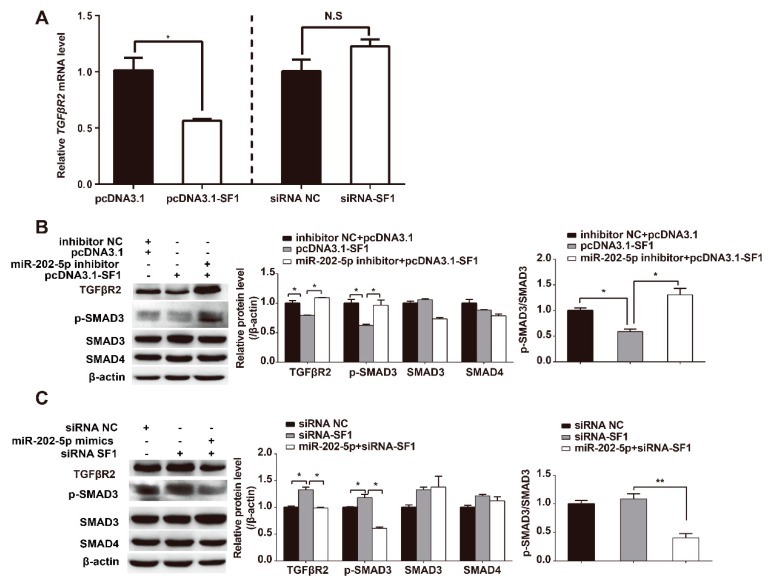
MiR-202-5p mediates the SF1 regulation of canonical TGF-β signaling pathway. (**A**) SF1 regulates TGFβR2 mRNA expression in GCs. Cells were transfected with pcDNA3.1-SF1 or SF1-siRNA. mRNAs were quantified by qPCR. (**B**) pcDNA3.1-SF1 and miR-202-5p inhibitors were cotransfected into GCs and protein levels were analyzed by WB. (**C**) SF1-siRNA and miR-202-5p mimics were cotransfected into GCs and protein levels were analyzed by WB. Experiments were repeated three times. Bars indicate the mean ± SEM of three independent replicates; **p* < 0.05, ***p* < 0.01, N.S.: Not significant.

**Figure 7 cells-09-00445-f007:**
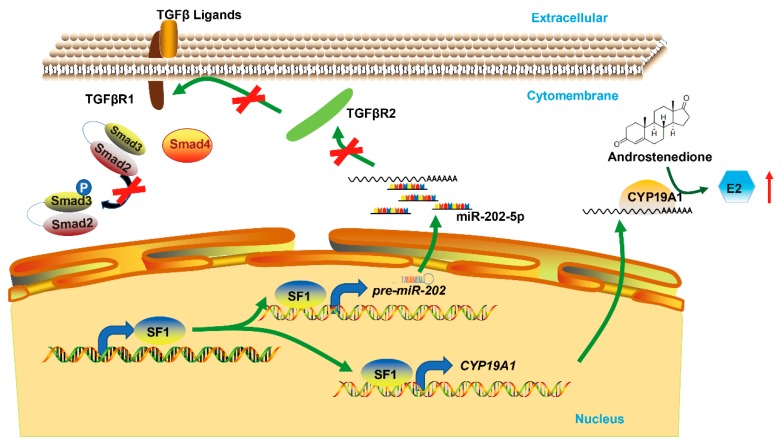
Schematic of miR-202-5p-mediated SF1 regulation of the canonical TGF-β signaling pathway in GCs.

## Data Availability

Qiang Ding had full access to all the data in this study and takes the responsibility for the integrity of the data and the accuracy of the data analysis. The data that support the findings of this study are available from the corresponding author upon reasonable request.

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
