# Peer review of "Transactivation of miR-202-5p by Steroidogenic Factor 1 (SF1) Induces Apoptosis in Goat Granulosa Cells by Targeting TGFβR2"

_cells, 2020, doi:10.3390/cells9020445_

Round 1
Reviewer 1 Report
In this manuscript the authors aim to demonstrate that miR-202-5P induces goat granulosa cells apoptosis by targeting TGFβR2. The study is comprehensive and most of the data are well presented. The major concern is that as shown in Figure 2 and others, the effects of miR-202-5P on apoptosis were minimal. How do the authors explain these results?
Author Response
在此手稿中,作者旨在证明miR-202-5P通过靶向TGFβR2诱导山羊颗粒细胞凋亡。这项研究是全面的,大多数数据都很好地呈现了出来。主要关注的是,如图2等所示,miR-202-5P对细胞凋亡的影响微乎其微。作者如何解释这些结果?
RE:在这项研究中,我们进行了生理测定以验证miR-202-5p在山羊颗粒细胞中的功能。在用miR-202-5p模拟物或抑制剂处理后,使用FACS,EdU分析,qPCR和Western blotting技术检测颗粒细胞的状态。所有处理均进行了三个独立的实验。根据原稿中显示的结果,可以暗示miR-202-5p功能性诱导GC凋亡。

Reviewer 2 Report
The paper by Qiang Ding et al. deals with the role of the microRNA 202-5p on TGF-beta signaling in the control of granulosa cell (GC) apoptosis. They show that miR-202-5p is specifically expressed in GCs of goat and induced apoptosis. They revealed that this microRNA targets the transforming growth factor-beta type II receptor (TGFβR2) for degradation and attenuated TGF-β/SMAD signaling by decreasing p-SMAD3 levels in GCs. Moreover, they showed that SF1 binds to the promoters of miR-202 and aromatase which resulted in miR-202-5p and CYP19A1 expression and estradiol (E2) release. Furthermore, SF1 repressed TGFβR2 and p-SMAD3 levels in GCs through the upregulation of miR-202-5p.
This article is clearly written and contains a series of novel and exciting observations. Principally, this work is suitable for publication in Cells after extensive language editing. However, I have a couple of points of critizism that need to be addressed by the authors before this work is acceptable for publication.
Major points
The authors analyzed the role of miR-202-5p in granulosa cells (GCs) which were derived from both small follicles and large follicles as well as from naked oocytes. However, in the sections involving transfection experiments it was not mentioned if the GCs used were from small or large follicles. This piece of information needs to be added. It would be interesting to perform the most crucial experiments in both GC types in a comparative fashion to reveal if there are differences in miR-202-5p function in GCs of different states of maturity. From the figure legends it is not clear if the authors have measured inter-assay variability or intra-assay variability. In order to confirm reproducibility of their data, statistical analysis must be done from calculating the means and SEM (or SD) from at least three independent experiments to assess inter-assay variability. This applies for Figures 1B+C, 2B-F, 3D-F, 4C+D, 5C+D, 6A-C. In the Methods section it must be clearly stated which statistical test has been applied (Student’s t-test?, unpaired?, two-tailed?).Minor points
Since the functional role of miR-202 in EVs has not been analyzed, this part should be condensed. In Fig. 2B+C expression of Bax and Bcl-2 needs to be measured by Western blot analysis to show if the changes in mRNA expression translate into corresponding changes in protein abundance. In Fig. 3B it is shown that there are at least 2 target binding sites for miR-202-5p in the goat TGFBR2 3’-UTR. However, in Fig.3C, it is not mentioned if both/all sites were mutated. It appears to me that only one site is mutated. If so, this one has to be specified. What about the functional relevance of the other binding sites? In Fig. 3E and Fig. 4D the authors did not observe an increase in protein levels of TGFBR2 and p-Smad3, respectively, upon treatment of GCs with the miR-202-5p inhibitor. How can this be explained? The authors should comment on this. The authors found that a miR-202-5p mimic suppressed proliferation. Given that this microRNA suppresses TGFBR2 expression and TGF-beta normally induces growth arrest in normal epithelial cells, this should have resulted in derepression of growth inhibition and hence an increase in proliferation. The authors should comment on this. In Fig. 4C and D the authors show strong signals for phospho-Smad3 in lysates of GCs although these cells have not been treated with exogenous TGF-beta. This suggests that TGF-beta-Smad signaling is constitutively active in GCs. How do the authors explain this observation and what is causing Smad3 activation. It is not immediately obvious to me why authors have measured estradiol secretion and aromatase expression, since there is no connection or regulatory interaction with miR-202-5p. It merely serves as a positive control for SF1 function. This is also evident from the scheme in Fig. 7 where no arrows are pointing from CYP19A1 or E2 towards miR-202-5p or TGFBR2. Lines 365-366: This sentence “TGFβR2 is regulated by miRNAs and causes apoptosis via …” should be rephrased. It is confusing since it reads as if this receptor is an inducer of apoptosis. Line 392-393. The authors mention that in human adrenocortical cells, TGF-β1 suppresses endogenous SF1 levels. Do the authors know whether this also the case in GCs? If so, they could speculate on the existence of a negative feedback loop involving TGFR2, SF1, miR-202-5p. Some abbreviations, i.e. SFF and LFF, are not identified in the text or the abbreviation section.
Author Response
对审阅者评论的回复:
丁强等人的论文。涉及微RNA 202-5p在TGF-β信号传导中对粒细胞(GC)凋亡的控制中的作用。他们表明,miR-202-5p在山羊GC中特异性表达,并诱导细胞凋亡。他们发现,该microRNA 通过降低GC中p-SMAD3的水平,靶向降解的转化生长因子βII型受体(TGFβR2)和减弱的TGF-β/ SMAD信号传导。此外,他们显示SF1与miR-202和芳香酶的启动子结合,导致miR-202-5p和 CYP19A1 表达以及雌二醇(E2)释放。此外,SF1通过上调miR-202-5p抑制了GC中的TGFβR2和p-SMAD3水平。
这篇文章写得清楚,包含一系列新颖而有趣的观察。原则上说,这项工作适合于 经过大量语言编辑的《 细胞》杂志上发表。但是,我有两点批评,在这项工作可以发表之前,作者需要解决。
要点
作者分析了miR-202-5p在颗粒细胞(GCs)中的作用,该颗粒细胞既有小卵泡又有大卵泡,还有裸卵母细胞。但是,在涉及转染实验的部分中,并未提及所使用的GC是来自小卵泡还是大卵泡。这条信息需要添加。以比较的方式在两种GC类型中执行最关键的实验,以揭示在不同成熟状态的GC中miR-202-5p功能是否存在差异,这将很有趣。
RE:好建议!显然,GC是在不同状态下从大小不同的卵泡中分离出来的。几项研究报告说,卵泡发育过程中卵巢中的GC转录组谱存在差异[1,2]。如“方法和材料”部分所述,我们选择了至少包含一个明显的大卵泡的卵巢,并选择了直径在2到5毫米之间的卵泡来分离GC。刺破并挤压这些卵泡以释放GC。我们在非血清培养基中体外培养了这种气相色谱,该培养基中含有卵泡刺激素(FSH)(1ng / ml)和其他补品。在这种情况下,我们不能确定从大卵泡或大卵泡中分离出的GC是否可以保持原始状态。因此,我们选择了GC的平衡条件来减少单元偏差。
从图例中,尚不清楚作者是否测量了批间差异或批内差异。为了确认其数据的可重复性,必须通过计算至少三个独立实验的均值和SEM(或SD)来进行统计分析,以评估批间差异。这适用于图1B + C,2B-F,3D-F,4C + D,5C + D,6A-C。在“方法”部分中,必须明确说明已应用了哪种统计检验(学生t检验,不成对,两尾?)。
RE:对于统计分析不清楚,我们深表歉意。我们已重新检查了本研究中的所有数据,以确保统计数据正确。我们在方法2.12和图形图例中添加了统计测试的描述。
Furthermore, we have polished our manuscript in an English Editing company and provided the certification.
Minor points
Since the functional role of miR-202 in EVs has not been analyzed, this part should be condensed.
RE: Thank you for your suggestion. We have condensed this part in the Discussion section.
In Fig. 2B+C expression of Bax and Bcl-2 needs to be measured by Western blot analysis to show if the changes in mRNA expression translate into corresponding changes in protein abundance.
RE: Thank reviewer remind us to add the WB analysis of BAX and Bcl-2. We added an additional Western blot figure in Figure S1 and described this result in the manuscript.
In Fig. 3B it is shown that there are at least 2 target binding sites for miR-202-5p in the goat TGFBR2 3’-UTR. However, in Fig.3C, it is not mentioned if both/all sites were mutated. It appears to me that only one site is mutated. If so, this one has to be specified. What about the functional relevance of the other binding sites?
RE: To confirm all miR-202-5p binding sites in goat TGFBR2 3’-UTR. We further mutated other two binding sites. The results showed that all three binding sites could reduce the luciferase activities when cells transfected with miR-202-5p mimics. We have added this part results in Fig.3C, 3D in manuscript.
In Fig. 3E and Fig. 4D the authors did not observe an increase in protein levels of TGFBR2 and p-Smad3, respectively, upon treatment of GCs with the miR-202-5p inhibitor. How can this be explained? The authors should comment on this.
RE: When GCs were cultured in vitro, the low level endogenous miR-202-5p possibly not significant affects protein levels of TGFBR2 and p-Smad3. So that, when GCs were treated with miR-202-5p inhibitor there were no obvious difference in protein levels of TGFBR2 and p-Smad3 compare to inhibitor NC. We have added in the Discussion section about this part.
The authors found that a miR-202-5p mimic suppressed proliferation. Given that this microRNA suppresses TGFBR2 expression and TGF-beta normally induces growth arrest in normal epithelial cells, this should have resulted in derepression of growth inhibition and hence an increase in proliferation. The authors should comment on this.
RE: Thanks for reviewer’s suggestion. We have added additional comments in the Discussion section. In fact, TGF-beta usually inhibits, but sometimes also stimulates, cell proliferation. In mammalian ovaries, accumulating evidence indicated that TGF-β participates in multiple ovary functions, including follicle growth[3] and GCs proliferation[4-6].
In Fig. 4C and D the authors show strong signals for phospho-Smad3 in lysates of GCs although these cells have not been treated with exogenous TGF-beta. This suggests that TGF-beta-Smad signaling is constitutively active in GCs. How do the authors explain this observation and what is causing Smad3 activation?
RE: Good point! Ovarian cells have been reported to produce three isoforms of the TGF-beta subfamily, namely TGF-β1, TGF-β2 and TGF-β3. Expression of TGF-β mRNA/protein in preantral follicles has been documented in several species including rodents, human, sheep and cattle[7-11]. And type-I and type-II TGF-β receptors amongst theca cells, granulosa cells and oocyte making TGF-Smad signaling is constitutively activity. For your question, we have read relevant references of TGF-Smad signaling. We noticed that Smad2 and 3 become phosphorylated and activated by several other ligands and receptor complexes of the TGFβ family [12] and a number of kinases can phosphorylate Smads, such as ERK1/2 MAP kinase[13]. In this study, we cultured the GCs in vitro with FSH which can constantly stimulate ERK1/2 MAP kinase activity. Furthermore, the phosphorylation levels of SMAD3 were also significantly increased by FSH, suggesting that TGF-β signaling is activated.
It is not immediately obvious to me why authors have measured estradiol secretion and aromatase expression, since there is no connection or regulatory interaction with miR-202-5p. It merely serves as a positive control for SF1 function. This is also evident from the scheme in Fig. 7 where no arrows are pointing from CYP19A1 or E2 towards miR-202-5p or TGFBR2.
RE:Thank reviewer’s suggestion. The relationship between miR-202-5p with estradiol secretion and aromatase expression was not in our consideration at first time. In the resubmitted manuscript, we have a positive result after re-statistic. The new results were added in Fig. 5A and 5B. we also added description in part 3.5 and commented this in the Discussion section. Here, we have used Spearman's rank correlation coefficient to evaluate the linear relation between genes expression and E2 concentrations.
Lines 365-366: This sentence “TGFβR2 is regulated by miRNAs and causes apoptosis via …” should be rephrased. It is confusing since it reads as if this receptor is an inducer of apoptosis.
RE:Thanks for reviewer’s suggestion, we have rephrased this sentence to make sure unambiguous.
Line 392-393. The authors mention that in human adrenocortical cells, TGF-β1 suppresses endogenous SF1 levels. Do the authors know whether this also the case in GCs? If so, they could speculate on the existence of a negative feedback loop involving TGFR2, SF1, miR-202-5p.
RE:如“讨论”部分所述,TGF-β1可以抑制人肾上腺皮质细胞[14]和Y1肾上腺肿瘤细胞[15]中的内源性SF1水平。然而,另一项研究报道,当大鼠卵巢GC中存在FSH时,TGF-β1可以增强SF1和LRH1(另一个核受体NR5A亚家族成员,肝受体同源物1)的表达[16]。它似乎由GC中miR-202-5p介导的SF1和TGF-β信号之间的负反馈回路组成。但是,需要进一步研究以揭示山羊GC中的这一现象。我们在“讨论”部分对此进行了评论。
在文本或缩写部分中未标识某些缩写,例如SFF和LFF。
RE:我们对此感到抱歉。我们已经检查了稿件中的所有缩写。

Reviewer 3 Report
The manuscript presented is far too verbose in the description of the experimental part. This means that the potential reader have to struggle (and not a little!) to find the right key to better understand the logic of the work and general conclusions. I therefore recommend a review of this work that takes into account these suggestions and may include a reflection on the extent that this experimental fact can assume in the problems related to women's pathologies.
Author Response
The manuscript presented is far too verbose in the description of the experimental part. This means that the potential reader have to struggle (and not a little!) to find the right key to better understand the logic of the work and general conclusions. I therefore recommend a review of this work that takes into account these suggestions and may include a reflection on the extent that this experimental fact can assume in the problems related to women's pathologies.
Re:Thank reviewer’s suggestion. We have re-structure the discussion of our manuscript to make sure the non-specialist can understand our research. We also consider our study could provide a suggestion to related to women's pathologies. Accelerated apoptosis of granulosa cells is one of the causes influencing follicle atresia and contributes to the etiology of polycystic ovarian syndrome (PCOS) in human ovaries. miR-202 may be a potential research target for the study of pathophysiological mechanisms of PCOS and other follicular atresia-related diseases.

Round 2
Reviewer 2 Report
The authors have satisfactorily responded to most points of my critique. However, I am not convinced that the data show inter-assay variability.
1. In their response they wrote: "We have added the description of statistical test in Methods 2.12 and figure legends." However, the phrasings "All samples were measured from at least three independent experiments", or "All experiments were treated with three replicates", or "Experiments were performed on a minimum of three occasions" do not state if the means of all three experiments are contained in the bars/data.
The sentence "Bars/data represent the mean ± SEM" may well mean that these are replicates from one single experiment. If the the authors have calculated the means and S.E.M. from three experiments then it should read: ""Bars (or data) represent the mean ± SEM (of the means) of three experiments."
The sentence in chapter 2.12. "All experiments were treated with three replicates." needs to be rephrased as it is unclear whether it refers to replicates/parallel samples within one single experiment or three independent experiments.
2. In their response to my minor point #6 they responded: "We noticed that Smad2 and 3 become phosphorylated and activated by several other ligands and receptor complexes of the TGFβ family [12] and a number of kinases can phosphorylate Smads, such as ERK1/2 MAP kinase [13]. In this study, we cultured the GCs in vitro with FSH which can constantly stimulate ERK1/2 MAP kinase activity."
This answer does not satisfy me since the authors have measured C-terminal phosphorylation of Smad3 (phospho-S423/S425). However, ERK1/2 can phosphorylate Smad3 and Smad2 only in their linker regions but NOT at the C-terminus! On this occasion the authors should clearly state in the text or in the figures 4 and 6 that C-terminal phosphorylation of Smad3 has been measured.
Author Response
The authors have satisfactorily responded to most points of my critique. However, I am not convinced that the data show inter-assay variability.
In their response they wrote: "We have added the description of statistical test in Methods 2.12 and figure legends." However, the phrasings "All samples were measured from at least three independent experiments", or "All experiments were treated with three replicates", or "Experiments were performed on a minimum of three occasions" do not state if the means of all three experiments are contained in the bars/data.The sentence "Bars/data represent the mean ± SEM" may well mean that these are replicates from one single experiment. If the authors have calculated the means and S.E.M. from three experiments then it should read: ""Bars (or data) represent the mean ± SEM (of the means) of three experiments."
The sentence in chapter 2.12. "All experiments were treated with three replicates." needs to be rephrased as it is unclear whether it refers to replicates/parallel samples within one single experiment or three independent experiments.
Re: Thank you minding us these confusing sentences. In this study, experiments for each group had three independent replicates. And each data of the means±SEM was calculated from three replicates. We have rephrased these sentences in chapter 2.12 and in figure legends. Here, we replaced the sentence “Bars/data represent the mean ± SEM” to “Bars indicate the mean ± SEM of three independent replicates”. And we rephrased the description in “Experiments for each group were repeated at least three times”
In their response to my minor point #6 they responded: "We noticed that Smad2 and 3 become phosphorylated and activated by several other ligands and receptor complexes of the TGFβ family [12] and a number of kinases can phosphorylate Smads, such as ERK1/2 MAP kinase [13]. In this study, we cultured the GCs in vitro with FSH which can constantly stimulate ERK1/2 MAP kinase activity."This answer does not satisfy me since the authors have measured C-terminal phosphorylation of Smad3 (phospho-S423/S425). However, ERK1/2 can phosphorylate Smad3 and Smad2 only in their linker regions but NOT at the C-terminus! On this occasion the authors should clearly state in the text or in the figures 4 and 6 that C-terminal phosphorylation of Smad3 has been measured.
Re: We are sorry about casting a suspicion on the phosphorylation of Smad3 activating by FSH. Recently studies demonstrated that FSH could phosphorylate SMAD3 at the C-terminal through TGFb signaling and FSHR/PKA signaling [1, 2]. So that, FSH stimulates the phosphorylation of SMAD3 both at linker region and C-terminal. We have pointed out that the phospho-SMAD3 was special at C-terminal phosphorylation (phospho-S423/S425) in results section. We have revised this part in the manuscript and added the comments at Discussion section.
References
Yang, J.; Zhang, Y.; Xu, X.; Li, J.; Yuan, F.; Bo, S.; Qiao, J.; Xia, G.; Su, Y.; Zhang, M. Transforming growth factor-β is involved in maintaining oocyte meiotic arrest by promoting natriuretic peptide type c expression in mouse granulosa cells. Cell Death & Disease 2019, 10, 558. Li, Y.; Jin, Y.; Liu, Y.; Shen, C.; Dong, J.; Xu, J. Smad3 regulates the diverse functions of rat granulosa cells relating to the fshr/pka signaling pathway. Reproduction 2013, 146, 169-179.